# The Importance of Wildlife Disease Monitoring as Part of Global Surveillance for Zoonotic Diseases: The Role of Australia

**DOI:** 10.3390/tropicalmed4010029

**Published:** 2019-02-06

**Authors:** Rupert Woods, Andrea Reiss, Keren Cox-Witton, Tiggy Grillo, Andrew Peters

**Affiliations:** 1Wildlife Health Australia, Mosman, NSW 2088, Australia; areiss@wildlifehealthaustralia.com.au (A.R.); kcox-witton@wildlifehealthaustralia.com.au (K.C.-W.); tgrillo@wildlifehealthaustralia.com.au (T.G.); 2World Organisation for Animal Health Working Group on Wildlife, 75017 Paris, France; 3School of Animal and Veterinary Sciences, E. H. Graham Centre for Agricultural Innovation, Charles Sturt University, Boorooma St., Wagga Wagga, New South Wales 2678, Australia; apeters@csu.edu.au

**Keywords:** Australia, emerging disease, international health regulations, Joint External Evaluation (JEE), One Health, Performance of Veterinary Services (PVS), surveillance, wildlife, zoonosis

## Abstract

Australia has a comprehensive system of capabilities and functions to prepare, detect and respond to health security threats. Strong cooperative links and coordination mechanisms exist between the human (public health) and animal arms of the health system in Australia. Wildlife is included in this system. Recent reviews of both the animal and human health sectors have highlighted Australia’s relative strengths in the detection and management of emerging zoonotic diseases. However, the risks to Australia posed by diseases with wildlife as part of their epidemiology will almost certainly become greater with changing land use and climate change and as societal attitudes bring wildlife, livestock and people into closer contact. These risks are not isolated to Australia but are global. A greater emphasis on wildlife disease surveillance to assist in the detection of emerging infectious diseases and integration of wildlife health into One Health policy will be critical in better preparing Australia and other countries in their efforts to recognize and manage the adverse impacts of zoonotic diseases on human health. Animal and human health practitioners are encouraged to consider wildlife in their day to day activities and to learn more about Australia’s system and how they can become more involved by visiting www.wildlifeheathaustralia.com.au.

## 1. Introduction

There is increasing recognition of the need to monitor as part of surveillance for emerging infectious diseases [1,2,3,4]. The majority of emerging infectious diseases are zoonoses with the predominant source shown to be wildlife [2,5]. Of specific concern is the impact and increase of wildlife-sourced zoonoses on human populations as globalisation, climate change and ecosystem alterations bring people and wildlife into closer contact. Importantly, many of the significant emerging infectious diseases in Australia have arisen in wildlife, and from within the country, rather than by overseas introductions, e.g., Hendra virus, Australian bat lyssavirus; see reviews [6,7]. For these reasons, Australia has implemented a general wildlife health surveillance system to enhance the early detection and characterization of microbial agents potentially involved with emerging diseases in free-ranging wildlife populations [6,7,8]. This paper briefly explains the governance for emerging zoonotic diseases and the roles played by non-human health professionals, especially those in the wildlife health sector in Australia. It concludes that though much good work has been done, there is an immediate need to improve integration of wildlife health into One Health policy as a critical step in better preparing Australia and other countries in their efforts to recognize and manage the adverse impacts of zoonotic diseases on human health.

## 2. Australia’s Biosecurity System and Wildlife Health Systems

Australia is a federation of six states and two territories. The public and animal health (production animals, domestic animals and wildlife) systems are complex, with a number of participants across the three level of government (Australian, state or territory and local) and in different sectors (human, animal and environment) [9,10,11]. The nationalised, broad-ranging animal health biosecurity system and the wildlife health component have been previously described [6,12,13]. Australia’s biosecurity system is complex, with activities carried out by Australian governments pre-border (offshore), border and post border (onshore) in collaboration with a large number of animal industry and other stakeholder groups, represented by a number of peak bodies. Under the Australian constitution, the Australian government is responsible for quarantine at the Australian border and also international animal health matters. State and territory governments are responsible for disease prevention, control and eradication within their boundaries. Preparedness plans and incident command structures adopted at both national and jurisdictional levels of government complement the system during emergency situations [11]. 

The framework ensures communication and cooperation between all levels of government and incorporates partnerships with animal industries and other stakeholders. An overarching National Animal Health Surveillance and Diagnostic Strategy Business Plan (Business Plan) guides investment in biosecurity priorities [14]. The wildlife component of the Business Plan focuses on nationally important and significant diseases of wildlife that may impact on Australia’s animal industries, human health, biodiversity, trade and tourism (‘wildlife diseases’). Emerging, exotic and zoonotic infectious diseases in addition to agriculturally significant diseases are emphasised.

Wildlife Health Australia (WHA) is a national body that works with Australian governments and stakeholders to improve preparedness, understanding and management of wildlife diseases. The current priority is the coordination of general surveillance and reporting of disease events in free-ranging wildlife. Over 30 surveillance partner agencies and organisations form the basis of Australia’s general wildlife health surveillance system, which includes Australian, state and territory government agriculture and environment agencies, 10 zoos, eight private veterinary hospitals and seven universities around Australia. A number of targeted national programs are also in place, including a Bat Health Focus Group and the National Avian Influenza Wild Bird Surveillance Program [13]. Biosecurity, health and environment professionals are included in all of these programs, thus providing strong linkage across sectors. Recognition of the role of non-government stakeholders and the use of a partnership-type approach is a strength of the system. 

A centralised, web-enabled national database of wildlife health information (‘eWHIS’) that is accessible across sectors by surveillance partners, both from within and outside of government, captures summary information on wildlife health and disease events submitted by surveillance partners in close to real time. About 40,000 wildlife cases are seen by WHA general surveillance partners each year [4] and one data category, ‘Interesting or Unusual’ wildlife cases, is designated to identify potential emerging infectious diseases. Within this category, between 200 and 300 ‘Interesting or Unusual’ wildlife cases are reported in Australia each year.

## 3. Australia’s Role in the Linkage and Coordination between Human and Animal Health Nationally and Internationally

Australia’s capability across animal and human health has recently been evaluated by international assessors utilizing the World Health Organization’s (WHO) Joint External Evaluation (JEE) against core capabilities and capacities under the International Health Regulations 2015, and the World Organisation for Animal Health’s (OIE) Performance of Veterinary Services (PVS) Evaluation [9,11]. It was concluded that Australia has a comprehensive system of capabilities and functions for preparedness, detection and response to health security threats. Australia’s system is strengthened through long-standing and stable cooperation links and coordination mechanisms that exist between the human and animal public health arms of the system [9]. The Australian Chief Veterinary Officer (ACVO) is Australia’s delegate to the OIE (OIE Delegate). There is a need for the ACVO and Australia’s state and territory Chief Veterinary Officers (CVOs) to maintain “line of sight” to the wildlife component of their animal health systems. Information provided by the wildlife surveillance system supports situation awareness and assessment of the risks posed by diseases of these animals. The ACVO coordinates Australia’s OIE work and draws on other specialists in Australian Government departments and agencies, industry bodies and other sources of expertise. Strong linkage exists between the ACVO and CVOs, Australia’s Chief Medical Officer and their respective departments [9]. Australia also has eight OIE Focal Points, focusing on specific animal-related topics such as wildlife, disease notification, communications and laboratories. These Focal Points support Australia’s OIE Delegate and provide linkages with their counterparts in other countries through the OIE network [9]. 

The recent PVS evaluation also highlighted Australia’s extraordinary commitment to biosecurity, serving the national interests by maintaining Australia’s high animal health status. The very high level of biosecurity within Australian animal health is founded on strong partnership collaboration and formal business arrangements amongst jurisdictions and with the private sector, including primary producers, processors, suppliers of inputs and laboratories. The PVS evaluation, which also included wildlife, emphasised Australia’s leadership role in the international veterinary community [11].

WHA supports the linkage and coordination between partners and across sectors by producing a regular electronic news digest that is distributed within Australia, to international members and regionally to OIE Wildlife Focal Points to share information on wildlife health and disease occurrences and issues of relevance to Australia and the region. Wildlife health surveillance data are summarised and publicly reported to the international community quarterly and yearly respectively via publications, namely Animal Health Surveillance Quarterly and Animal Health in Australia [15]. Summaries are also provided to OIE at six-monthly intervals, and WHA produces a six-month summary of Australian bat lyssavirus general surveillance data as “Bat Stats” [16]. Where possible, Australia’s wildlife health data are also provided to open source databases (for example, the provision of sequence data generated by the Avian Influenza Wild Bird Surveillance Program to GenBank) and to help satisfy international reporting requirements (see [15,17]). Other relevant outputs coordinated by WHA include fact sheets and national biosecurity guidelines for wildlife health.

The processes in place for information capture, provision and reporting allow rapid and timely submission of wildlife health and disease information to the national system, assessment and notification to the relevant authorities. 

## 4. Key Challenges and Opportunities Identified from the Australian Experience

Arguably the key challenge and opportunity emerging from Australia’s experience in wildlife disease monitoring as part of surveillance for emerging infectious diseases is the difficulty in finding objective indicators of success [18]. Surveillance and response systems face considerable subjectivity if measurable outcomes, assessment and improvement of the efficacy and efficiency of wildlife disease preparedness remain lacking. Objectives for Australia’s general wildlife health surveillance system are to: Improve Australia’s ability to describe the occurrence and distribution of wildlife diseases.Allow early detection of unusual wildlife disease events including changes in the pattern of existing diseases and occurrence of emerging or exotic diseases.Provide basic data that is able to support more detailed *ad hoc* disease investigations.Provide data to support claims of freedom from specified diseases and answer queries from trading partners as requested.Identify and capture all sources of animal health information that would effectively contribute to Australia’s overall understanding of its wildlife health.Remain highly cost effective and maximise the representativeness and coverage of the system.Improve and expand the capacity to collect information about feral animals, especially from non-government sources.

Seeking stakeholders’ and users’ input on how data and services provided by WHA improve their ability to identify and manage wildlife disease risk and preparedness is a good example of the pragmatic approach to measure success that is currently used. Effective detection and eradication of emerging infectious diseases in wildlife requires the collection of objective evidence to demonstrate the robustness of wildlife disease monitoring systems. This information, in turn, will guide adequate deployment of resources and implementation of system improvements. We are not aware of any examples of emerging infectious diseases of Australian native wildlife that have been eradicated or locally eliminated by Australian state or territory governments. A discussion of tools and tactics is beyond the scope of this opinion piece but includes options routinely deployed in production, invasive and feral animal response. Though local elimination and or proof of eradication appear to be conceptually simple measures of success, a common indicator used in human and animal health economics, economic loss averted (ELA), could also be deployed. Despite some of the challenges, the use of ELA would not only allow a greater understanding of the benefit cost of the system, but also allow comparisons to be made with other national animal and human health risk mitigation programs. 

Wildlife disease surveillance faces other recognised challenges. There is incomplete knowledge of wildlife population demographics and distributions, as well as legitimate questions of surveillance sensitivity and potential biases in results. Australia uses a pragmatic approach, focusing on the development of a good general surveillance system, rapid reporting, as well as the identification and investigation of clusters of wildlife deaths or morbidity. The supporting architecture for the wildlife health surveillance system is based on Australia’s livestock biosecurity framework and has historically focussed on the capture and provision of information to support trade and market access. Recognising that a general surveillance system to support one sector also supports others, a greater focus on zoonotic diseases and diseases which may impact wildlife populations and biodiversity and their inclusion in general surveillance activities would significantly strengthen the Australian system. The recent work of Craik, Palmer and Sheldrake [10] concluded that there was an immediate need to further invest in environmental biosecurity and bring it more fully into mainstream biosecurity activities in Australia. The inclusion of the environmental sector in arrangements targeting wildlife health and diseases in Australia would significantly improve the ability to detect new and emerging diseases with the potential to impact upon animal and or human health. The recent appointment of a Chief Environmental Biosecurity Officer for Australia (ACEBO) is a significant development that offers opportunities in improving the coordination and linkage between environment, health and agriculture.

More broadly, there remains significant opportunity for improvement of the position of wildlife within Australia’s wider biosecurity arrangements. There are challenges, however, with maintaining a high operational functionality in Australia’s complex system. The JEE concluded that despite outstanding progress in developing and implementing steps to ensure a collaborative approach between the human and animal health sectors, opportunities remain for the development of greater coordination of activities [9]. Given the risks posed by anthropogenic changes that have the potential to spark disease outbreaks in wildlife populations and the potential emergence of zoonotic diseases, each of the observations proposed by the JEE could be enhanced by the further consideration and inclusion of wildlife and environmental health: Development of an all-hazards health protection framework. The national framework for communicable disease control could be further developed with an increased emphasis on the risks posed by anthropogenic changes to the environment, which are linked to disease emergence in wildlife, changes in relative distribution and composition of infectious agents and species affected.Public and animal health workforce issues. Some specific competencies were recognized for which there is a limited workforce and future replacement may be at risk. For wildlife, this includes disease ecologists and disease and wildlife emergency response managers. Australia’s PVS evaluation noted that in several jurisdictions staff levels are seen to be severely inadequate [11]. Increased investment in on-the-ground veterinary officer deployment for investigation and surveillance activities is required. Only some of Australia’s environmental agencies include veterinarians and a placement within each of these agencies would also facilitate communication and linkage with counterparts in agriculture and public health agencies.The use of genomic data in disease surveillance, which could be better harnessed for pathogen discovery, surveillance work and elucidating the epidemiology at population interfaces, for example, at the wildlife–human and wildlife–livestock interfaces. A sequence data management and interpretation framework bridging bio-informatics and evolutionary microbiology (phylogenetics, phylogeography) is of critical importance in comprehensively holistic programs, particularly to provide an adequate ecological and evolutionary interpretation of the relationships between agents discovered in wildlife and zoonotic agents affecting human populations. All of this has the ultimate purpose of tracing the potential origins of zoonotic diseases, unveiling mechanisms as to how wildlife-associated agents may break cross species transmission barriers (host shifts) or simply quantifying and qualifying transient cross-species spillover infections. The European COMPARE project is an example of an effective approach to tackling emerging infectious diseases, ranging from risk assessment, sampling frames and surveillance, application of new generation sequencing, and data flow into databases, to the development of harmonized approaches across human, livestock and wildlife populations [19].Joint training and emergency animal disease response exercises across Australian Government agencies, relevant state and territory government agencies, and wildlife stakeholders, along with strategic risk assessment of current preparedness activities and arrangements for wildlife, would help to identify areas requiring improvement.Wildlife monitoring also presents an opportunity to assist with linkage across sectors in the areas of surveillance, preparedness and investigation. However, simpler management structures are required and the use of WHA, a public-private partnership built on One Health principles, to assist as a “trusted broker” represents a potential opportunity for the system that needs to be further developed.Information technology and mapping systems between Australian jurisdictions are not yet fully compatible [11]. Linkage of jurisdictional information systems to the eWHIS would remove redundancy, improve efficiency and allow analysis at a whole of country scale.

Following the completion of a JEE, the WHO recommends that countries develop a National Action Plan for Health Security (NAPHS) to address the recommendations in the JEE Mission Report. In keeping with the JEE ideology, the NAPHS is developed collaboratively across multiples sectors, with the aim of prioritising the implementation of recommendations to improve compliance with international health regulations and national health security. Specific recommendations for zoonotic diseases in Australia’s NAPHS are: “Introduce a formal process through committee structures between human health and animal health to regularly review a joint list of priority zoonotic diseases. Consider designating zoonotic diseases of public health importance in Australia as nationally notifiable in animals.Establish a dedicated multisectoral national zoonosis committee or ensure reciprocal animal and human sector representation on their respective national zoonotic disease-related committees to enhance communications, bridge knowledge gaps and strengthen collaborative responses.Consider standardising/aligning laboratory case definitions and typing between human and animal health sectors to enhance data comparison of their surveillance systems [20]”.

In addressing these recommendations, it is important for Australia that emerging infectious diseases and zoonoses of wildlife be included.

## 5. Conclusions

The risks to Australia posed by wildlife diseases will almost certainly become greater with anthropogenic changes such as a climate change, changes in land use, as well as societal attitudes that bring wildlife, livestock and people into closer contact [6]. The challenges of emerging infectious diseases from wildlife is, however, a global issue. A greater emphasis on wildlife disease surveillance to assist in the detection of emerging infectious diseases and integration of wildlife and environmental health into One Health policy will be critical in better preparing Australia and other countries in their efforts to recognize and manage the adverse impacts of zoonotic diseases on human health. Animal and human health professionals, including those in the community, are reminded of Australia’s system of arrangements for wildlife health and are encouraged to consider wildlife health in their practices. More information on Australia’s system and how they can become involved and contribute to improving the integration of wildlife health into their practice and communicate within an evolving network of partners is available at www.wildlifehealthaustralia.com.au.

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
