# Peer review of "The Importance of Wildlife Disease Monitoring as Part of Global Surveillance for Zoonotic Diseases: The Role of Australia"

_tropicalmed, 2019, doi:10.3390/tropicalmed4010029_

Round 1

Reviewer 1 Report

Incorporation of wildlife are critical to a basic understanding emerging infectious disease issues, particularly in an island ecosystem. Australia is no exception, especially as many diseases are absent from this continent. On this subject, the authors present their opinions in a One Health context to educate readers unfamiliar with this unique ecosystem. As such, overall the title is descriptive, the approach is sound, the examples are relevant, the citations are supportive and the paper is generally well written.

Ideally, to break up the text, it would be nice to include a relevant table, figure and photo to help augment the MS and to better capture reader attention.

Minor comments follow.

Ln 59 What disease examples have been eradicated versus local elimination, by state and local governments?

Ln 131-141 Be consistent in adding periods or not at the end of the bullet points and elsewhere in the document.

Ln 145 What wildlife EIDs have been eradicated and how would this proceed in the future once detected – by culling?

Author Response

Thanks for the positive feedback.  All are good suggestions that will improve the work: thank you.

...nice to include a relevant table, figure and photo to help augment the MS and to better capture reader attention...

It's a great idea to include a relevant table, figure and photo, however it starts to expand the work out into a more comprehensive paper (we would need to explain and discuss the table and figure and we were asked to provide something of about a thousand words).  I'm suggesting therefore that rather than including the three we go with an image and will include one with the resubmitted manuscript.  It does suggest to me, however, that something more in-depth would be useful for this journal in future: thank you (as do your following comments as well).

Ln 59 What disease examples have been eradicated versus local elimination, by state and local governments?

We are not aware of any examples of EIDs of wildlife that have been eradicated versus local elimination and have included this in the MS in the section that discusses effectiveness of the wildlife surveillance system (Ln 168 onwards on the updated version).

Ln 131-141 Be consistent in adding periods or not at the end of the bullet points and elsewhere in the document.

We've addressed this through the document (thank you) and have checked with the government style guide and our in-house editor here regarding use in lists.  Her advice was that: "The way this list is punctuated currently is correct, full stop at end. Not full sentences in this list. Other lists are full sentences so can have a full stop at end of each point."  I have therefore left the list as it is but have edited else where.  I hope this is OK.

Ln 145 What wildlife EIDs have been eradicated and how would this proceed in the future once detected – by culling?

I've combined some comments from Ln 168 onwards.  We've just written guidelines for the management of an emergency disease in wildlife, which presents an approach and tools and tactics.  It's a big topic but basically everything is in scope.  The social license issues are, as you can imagine, considerable however! 

Reviewer 2 Report

The opinion article entitled “The importance of Wildlife disease monitoring as part of the global surveillance for zoonotic diseases: the role of Australia”, is a timely piece that argues on the critical importance of including the monitoring of microbial agents causing disease in wildlife populations, as a means to more effectively anticipate disease detection, preparedness and control of potential zoonotic diseases. Authors address such an interesting topic by fully describing the current framework set up by Australia’s government to resolve ongoing issues linked to food safety, public health and conservation of the indigenous natural bio-diversity. It is interesting that the original multidisciplinary, multi-sectorial, multi-institutional scope adopted by Australia, meant to integrate animal and public health in a sort of “One Health” umbrella. However, authors stressed that since such a system was originally designed to deal with exotic threats rather than with ingenious ones. The supporting architecture of the program completely dismissed the inclusion of diseases affecting wildlife, which at some point also can affect public and animal health, as well as animal-derived food safety (they cited the examples of NIPA and HENDRA viruses). In their piece, authors revealed this critical gap exist in current global “One Health” policies (very assertive point), suggesting it should be amended by including a wildlife health component in it. In a very pragmatic way, authors suggest using the existing integral framework, which includes multidisciplinary, multisectorial, multi-institutional communication and collaboration networks, by only integrating professionals and institutions dealing with wildlife health in a daily basis (which most countries already have).

Major suggestions.

Recognizing that wildlife health is intimately intertwined with the health of the environment, why not to suggest integrating this other component(environmental health) to engender a more complete One Health perspective, which not only monitors, detects and mitigates disease emergence in wildlife, but also investigates the real causes of it.

Infectious microbial agents found in wildlife often are not involved with disease in their natural host populations. Thus, a better understanding of host-microbial organisms relationships (understanding all types or symbiotic relationships and how they may be altered over time in response to anthropogenic changes) should be integrated in any suggested framework in order to provide more comprehensive inferences on their potential roles as agents of emerging diseases in other animal species including humans.

Related to the former point and in regards the use of next generation sequence data as a means for identifying and characterizing microbial organisms found in wildlife and in their environments. It is highly important to stress that a huge body of knowledge needs to be generated to better understand how we can use whole genome nucleotide sequence data and phylogenetic reconstructions to detect, infer, and predict how organisms sharing a recent or distant ancestry with currently known human zoonotic pathogens could have evolved into human or other animal pathogens. What would it be the ecological (in a more ample context, ie., environment, host) mechanisms leading to favoring host jumping and later leading pathogenicity in their new hosts (or in their primary hosts). Having a better understanding on how microbial agents evolve in pristine, versus human-intervened or altered environments may help to better anticipate disease emergence. This demands other fields of expertise beyond veterinarians, physicians and disease ecologists, it requires, bio-informaticians, spatial modelers, microbiologists, microbial evolution experts to be able to integrate such a knowledge in a more holistic precise way. Particularly, to avoid misleading interpretation of the data, which can end up in measures driving the extinction of some wildlife species or damaging more the existing global bio-diversity.

Minor detailed suggestions.

Although, probably out of the scope of this piece, authors probably should elaborate more why they feel convenient leaving environmental health out of the equation, for the moment. I should probably argue that several countries around the world already have federal and privately funded agencies dealing with issues directly or indirectly involved with the health of the environment.

I think it would make a lot of sense adding an operational definition for animal health, since it clearly only involves matters concerning the health of domestic and domesticated animals that are part of food production industry.

Sentence running from lines 40 to 42. Please make it active. My suggestion for rephrasing it would be, "For this reasons, Australia has implemented a general wildlife health surveillance system to enhance early detection and characterization of microbial agents potentially involved with emerging infectious diseases in free ranging wildlife populations". I think surveillance primarily means to detect and characterize the microbial agents, then we assess their potential involvement with emerging diseases..

Line 52. I think that should be and “or” instead of “and”, for it is a territory or a State

Line 58 I would suggest to change “territorial” rather than territory. Territorial government as opposed to territory government.

Line 66 “of” instead of “with” and remove “as part of their epidemiology” since it cuts flow to the sentence and does not add any meaning.

Line 70, remove marked “and” and unnecessary commas..

Line 73. Is there only one zoo or a network of zoos involved?? Please clarify. In the same line, please clarify whether zoo or zoos are part of the National Avian Influenza Wild Bird program.

Line 74. Is there only one Sentinel Clinic and University involved or it is a network of sentinel clinics and Universities.

Line 82. I think “designated” fits better than “designed”

Line 92. For preparedness, detection and response.

Line 93. Long standing and stable instead of “strong”

Line 95, who relies on the expertise…

Line 97. and other specialists instead of expertise.

Line 107 , which also included wildlife.

Line 114, use namely, to link the sentence as follows “ via publications namely, Animal Health Surveillance Quaterly…..

Line 127. Measures … “indicators”

Lines 127 to 129. Please rephrase the entire sentence to improve clarity. Suggestion.. “Surveillance and response systems face considerable subjectivity if measurable outcomes, assessment and improvement of the efficacy and efficiency of wildlife disease preparedness remain lacking”.

Paragraph running from lines 142 to 144, should be rephrased in active voice to increase clarity and should include aspects on health economics such as, “economical loss averted” for early interventions due to early detection of EIDs. This is a common indicator used in human and animal health economics.

Suggestion “Seeking stakeholders and users input on how data and services provided by WHA improve their ability to identify and manage wildlife disease risk and preparedness, would be a good example of a pragmatic approach to measure success that is currently used”.

Paragraph running from line 145 to 147. Please rephrase the entire paragraph using active voice to improve clarity.

Suggestion. “Effective detection and eradication of emerging infectious diseases in wildlife requires the recollection of objective evidence to demonstrate the robustness of wildlife disease monitoring systems. This information, in turn, will guide adequate deployment of resources and implementation of system improvements”.

Line 148. Substitute “has”  for “face”

Lines 149 to 150. Please modify marked sentence for “, as well as there are legitimate questions on surveillance sensitivity and potential biases in results.”

Lines 150 to 151. On the second sentence. I find pertinent to use active voice and a connector. Suggestion:

In this regard, Australia used another pragmatic approach again by focusing on………

Line 151, substitute the marked “and” for “as well as”

Line 152. To eliminate repetition I would suggest start the sentence with “ The supporting architecture for the wildlife health surveillance system….”

Lines 154 and 155. The paragraph.. Recognizing that a general surveillance system created to support one sector may also support others, the integration of a broader program focused on diseases (including zoonotic) that may affect the health and biodiversity of wildlife populations would significantly strengthen the existing Australian system.

Line 170. I would suggest to rephrase this entire sentence, for it is not wildlife that poses the risk, but the anthropogenic changes that disrupt the natural balance in the environment (truly, the one health concept) are the ones posing the risks on wildlife populations, which ignite the emergence of disease and zoonotic diseases. Thus, the inclusion of a wildlife component will definitively help to unveil the consequences of such imbalance, but not the root causes.

Suggestion: “ Given the risks posed by anthropogenic changes that are the ones that ignite disease in wildlife populations and potential emergence of zoonotic diseases, each of the observation proposed by the JEE……….inclusion of wildlife and environmental health.

Line 173. The point on the “Development of an all-hazards health protection framework. The national framework for communicable disease control could be further developed with an increased emphasis on the risks posed by anthropogenic changes in the environment, which are linked to disease emergence in wildlife, changes in relative distribution and composition of infectious agents and species affected.

Line 218. A sequence data management and interpretation framework bridging bio-informatics and evolutionary microbiology (phylogentics, phylogeography) is of critical important in comprehensively holistic programs. Particularly, to provide an adequate ecological and evolutionary interpretation of the relationships between agents discovered in wildlife and zoonotic agents affecting human populations. All this with the ultimate purpose of tracing the potential origins of zoonotic diseases, unveiled mechanisms as to how wildlife-associated agents may break cross species transmission barriers (hosts shifts) or simply to quantify and qualify transient cross-species spillover infections.

Line 221. Sentence starting at line 220. Suggested change. The risks…….will almost certainly..with anthropogenic changes such as, climate change, changes in land used, as well as societal attitudes that bring wildlife, livestock and people into closer contact.

Line 223. In sentence starting with “ A greater emphasis..and integration of wildlife and environmental health into One Health policies…..

Author Response

These are terrifying suggestions: thank you.  We've made the minor editorial changes as suggested and I've tried to work in your other suggestions where indicated (they have greatly improved the article).  I've also included some replies to your specific comments below as well.  Thank you again and Best Wishes, Rupe Woods.

The opinion article entitled “The importance of Wildlife disease monitoring as part of the global surveillance for zoonotic diseases: the role of Australia”, is a timely piece that argues on the critical importance of including the monitoring of microbial agents causing disease in wildlife populations, as a means to more effectively anticipate disease detection, preparedness and control of potential zoonotic diseases. Authors address such an interesting topic by fully describing the current framework set up by Australia’s government to resolve ongoing issues linked to food safety, public health and conservation of the indigenous natural bio-diversity. It is interesting that the original multidisciplinary, multi-sectorial, multi-institutional scope adopted by Australia, meant to integrate animal and public health in a sort of “One Health” umbrella. However, authors stressed that since such a system was originally designed to deal with exotic threats rather than with ingenious ones. The supporting architecture of the program completely dismissed the inclusion of diseases affecting wildlife, which at some point also can affect public and animal health, as well as animal-derived food safety (they cited the examples of NIPA and HENDRA viruses). In their piece, authors revealed this critical gap exist in current global “One Health” policies (very assertive point), suggesting it should be amended by including a wildlife health component in it. In a very pragmatic way, authors suggest using the existing integral framework, which includes multidisciplinary, multisectorial, multi-institutional communication and collaboration networks, by only integrating professionals and institutions dealing with wildlife health in a daily basis (which most countries already have).

Beautifully put, you would have been a much better person to write this than us!  It was a bit of a late request and we were scrambling to try to help (we aren't scientists either and have just done the best we could...).

Major suggestions.

Recognizing that wildlife health is intimately intertwined with the health of the environment, why not to suggest integrating this other component(environmental health) to engender a more complete One Health perspective, which not only monitors, detects and mitigates disease emergence in wildlife, but also investigates the real causes of it.

Infectious microbial agents found in wildlife often are not involved with disease in their natural host populations. Thus, a better understanding of host-microbial organisms relationships (understanding all types or symbiotic relationships and how they may be altered over time in response to anthropogenic changes) should be integrated in any suggested framework in order to provide more comprehensive inferences on their potential roles as agents of emerging diseases in other animal species including humans.

Related to the former point and in regards the use of next generation sequence data as a means for identifying and characterizing microbial organisms found in wildlife and in their environments. It is highly important to stress that a huge body of knowledge needs to be generated to better understand how we can use whole genome nucleotide sequence data and phylogenetic reconstructions to detect, infer, and predict how organisms sharing a recent or distant ancestry with currently known human zoonotic pathogens could have evolved into human or other animal pathogens. What would it be the ecological (in a more ample context, ie., environment, host) mechanisms leading to favoring host jumping and later leading pathogenicity in their new hosts (or in their primary hosts). Having a better understanding on how microbial agents evolve in pristine, versus human-intervened or altered environments may help to better anticipate disease emergence. This demands other fields of expertise beyond veterinarians, physicians and disease ecologists, it requires, bio-informaticians, spatial modelers, microbiologists, microbial evolution experts to be able to integrate such a knowledge in a more holistic precise way. Particularly, to avoid misleading interpretation of the data, which can end up in measures driving the extinction of some wildlife species or damaging more the existing global bio-diversity.

Agree and we have tried to incorporate this more forcefully in the MS now (your comments and suggestions below help and we have incorporated those and or made the changes you suggest).  Our initial brief was to provide a thousand words and to focus on the human and animal system ("two thirds health"), so it is great to get this feedback and be able to be a bit more assertive with this.  We always feel like the forgotten part in all the One Health stuff, so it is wonderful to hear from someone who really seems to "get it".  Thank you. 

 Minor detailed suggestions.

Although, probably out of the scope of this piece, authors probably should elaborate more why they feel convenient leaving environmental health out of the equation, for the moment. I should probably argue that several countries around the world already have federal and privately funded agencies dealing with issues directly or indirectly involved with the health of the environment.

Reply: See comment above :) 

I think it would make a lot of sense adding an operational definition for animal health, since it clearly only involves matters concerning the health of domestic and domesticated animals that are part of food production industry.

Yes, agree.  However, this has proven very difficult for us.  My understanding is that Australia does not have an agreed definition.  The guidance I have sought from some fairly senior people (who will remain nameless) is "Can you imagine how hard this would be to reach consensus in our system?".  Even OIE seems to struggle with this.  It has something close "including both domestic animals and wildlife", which I have tried to include as, I think that is the intention i.e. to determine if wildlife are in or out of scope.

Sentence running from lines 40 to 42. Please make it active. My suggestion for rephrasing it would be, "For this reasons, Australia has implemented a general wildlife health surveillance system to enhance early detection and characterization of microbial agents potentially involved with emerging infectious diseases in free ranging wildlife populations". I think surveillance primarily means to detect and characterize the microbial agents, then we assess their potential involvement with emerging diseases.

Done: thank you.

Line 52. I think that should be and “or” instead of “and”, for it is a territory or a State

Done: thank you.

Line 58 I would suggest to change “territorial” rather than territory. Territorial government as opposed to territory government.

I agree that it sounds better, but this is the standard terminology that is used in Australia so have left it as it is.  I suggest that we let the section editors make a decision on this.

Line 66 “of” instead of “with” and remove “as part of their epidemiology” since it cuts flow to the sentence and does not add any meaning.

Done.

Line 70, remove marked “and” and unnecessary commas..

Done.

Line 73. Is there only one zoo or a network of zoos involved?? Please clarify. In the same line, please clarify whether zoo or zoos are part of the National Avian Influenza Wild Bird program.

Done.  More detail added.

Line 74. Is there only one Sentinel Clinic and University involved or it is a network of sentinel clinics and Universities.

More detail on the program added.

Line 82. I think “designated” fits better than “designed”

Change made.

Line 92. For preparedness, detection and response.

Changes made.

Line 93. Long standing and stable instead of “strong”

Change made.

Line 95, who relies on the expertise…

Two sentences added to explain, and also briefly the main uses of the information.

Line 97. and other specialists instead of expertise.

Change made.  Thank you.

Line 107 , which also included wildlife.

Change made.

Line 114, use namely, to link the sentence as follows “ via publications namely, Animal Health Surveillance Quaterly…..

Change made.

Line 127. Measures … “indicators”

Change made.

Lines 127 to 129. Please rephrase the entire sentence to improve clarity. Suggestion.. “Surveillance and response systems face considerable subjectivity if measurable outcomes, assessment and improvement of the efficacy and efficiency of wildlife disease preparedness remain lacking”.

Done.  Thank you.

Paragraph running from lines 142 to 144, should be rephrased in active voice to increase clarity and should include aspects on health economics such as, “economical loss averted” for early interventions due to early detection of EIDs. This is a common indicator used in human and animal health economics.

Have tried to do this as suggested and have incorporate a brief statement on ELA.

Suggestion “Seeking stakeholders and users input on how data and services provided by WHA improve their ability to identify and manage wildlife disease risk and preparedness, would be a good example of a pragmatic approach to measure success that is currently used”.

Done.

Paragraph running from line 145 to 147. Please rephrase the entire paragraph using active voice to improve clarity.

Suggestion. “Effective detection and eradication of emerging infectious diseases in wildlife requires the recollection of objective evidence to demonstrate the robustness of wildlife disease monitoring systems. This information, in turn, will guide adequate deployment of resources and implementation of system improvements”.

Done.

Line 148. Substitute “has”  for “face”

Done

Lines 149 to 150. Please modify marked sentence for “, as well as there are legitimate questions on surveillance sensitivity and potential biases in results.”

Done

Lines 150 to 151. On the second sentence. I find pertinent to use active voice and a connector. Suggestion:

In this regard, Australia used another pragmatic approach again by focusing on………

Done

Line 151, substitute the marked “and” for “as well as”

Done

Line 152. To eliminate repetition I would suggest start the sentence with “ The supporting architecture for the wildlife health surveillance system….”

Done

Lines 154 and 155. The paragraph.. Recognizing that a general surveillance system created to support one sector may also support others, the integration of a broader program focused on diseases (including zoonotic) that may affect the health and biodiversity of wildlife populations would significantly strengthen the existing Australian system.

Done

Line 170. I would suggest to rephrase this entire sentence, for it is not wildlife that poses the risk, but the anthropogenic changes that disrupt the natural balance in the environment (truly, the one health concept) are the ones posing the risks on wildlife populations, which ignite the emergence of disease and zoonotic diseases. Thus, the inclusion of a wildlife component will definitively help to unveil the consequences of such imbalance, but not the root causes.

Again, beautifully put!  Changes made (below).

Suggestion: “ Given the risks posed by anthropogenic changes that are the ones that ignite disease in wildlife populations and potential emergence of zoonotic diseases, each of the observation proposed by the JEE……….inclusion of wildlife and environmental health.

Changes made.

Line 173. The point on the “Development of an all-hazards health protection framework. The national framework for communicable disease control could be further developed with an increased emphasis on the risks posed by anthropogenic changes in the environment, which are linked to disease emergence in wildlife, changes in relative distribution and composition of infectious agents and species affected.

Done

Line 218. A sequence data management and interpretation framework bridging bio-informatics and evolutionary microbiology (phylogentics, phylogeography) is of critical important in comprehensively holistic programs. Particularly, to provide an adequate ecological and evolutionary interpretation of the relationships between agents discovered in wildlife and zoonotic agents affecting human populations. All this with the ultimate purpose of tracing the potential origins of zoonotic diseases, unveiled mechanisms as to how wildlife-associated agents may break cross species transmission barriers (hosts shifts) or simply to quantify and qualify transient cross-species spillover infections. 

Again, beautifully put.

Line 221. Sentence starting at line 220. Suggested change. The risks…….will almost certainly..with anthropogenic changes such as, climate change, changes in land used, as well as societal attitudes that bring wildlife, livestock and people into closer contact.

Changes made

Line 223. In sentence starting with “ A greater emphasis..and integration of wildlife and environmental health into One Health policies…..

Changes made

Wow, if this was anywhere nearly as tiring reviewing as it was making the changes then I owe you big time.  Running out of steam a bit but I think we got there.  Again thank you...I've given up on trying to keep to the requested 1000 words, but I've learnt a lot and the work is so much more valuable with your insights.  It doesn't look like you want your name released, but I have added you (and the other referees) to the acknowledgements.  I hope that is enough.

Reviewer 3 Report

Most of public health laboratories have conducted predominantly surveillance for critical infectious diseases, such as influenza, rabies in animals, and arboviruses. As new infectious agents emerge and older ones alter their patterns of distribution, these laboratories may have to employ new strategies and tactics for disease surveillance. Among these are zoonotic diseases being neglected in many countries. We urgently need strategies allowing to control and prevent new infections. Surveillance strategies must be sufficiently flexible to adapt to the circumstances that we recognize as a disease emerges. With zoonoses, the emergence may be complete before detectable disease occurs in humans. This provides an opportunity to anticipate disease in a population and prevent its appearance by implementing appropriate control measures. The ecosystems are complex, however, and we must adapt our surveillance strategies as we learn more about the pathogens, their vectors, and the reservoirs. Temporal and spatial relationships should cause us to change our strategies as the disease advances. Different strategies should apply in areas where emergence is complete than in areas where disease is emerging or yet unknown. The role of Austraia, continent-country, and zoonotic potential of animals living here is underestimated. Taking into account increasing numer of zoonotic infections in humans we need to take care about OneHealth approach. Therefore opinions presented by Autors are waluable and should be taken into consideration by appropriate agenicies nd services.

I have no specific comments and/or suggestion to authors.

Author Response

Thank you for your insightful comments and for taking the time to read and consider this work.  I am very grateful and hopefully we can meet sometime in future and or we can help you out as well somewhere along the journey.

Rupe Woods